# Phenolic-Rich Indian Almond (*Terminalia catappa* Linn) Leaf Extract Ameliorates Lipid Metabolism and Inflammation in High-Fat Diet (HFD)-Induced Obese Mice

**DOI:** 10.3390/metabo15090594

**Published:** 2025-09-08

**Authors:** Opeyemi O. Deji-Oloruntoba, Ji Eun Kim, Hee Jin Song, Ayun Seol, Dae Youn Hwang, Miran Jang

**Affiliations:** 1Biohealth Convergence Unit, Food, and Drug Biotechnology, Inje University, Gimhae 50834, Republic of Korea; oodejioloruntoba@gmail.com; 2Department of Biomaterials Science (BK21 FOUR Program), Life and Industry Convergence Research Institute, College of Natural Resources and Life Science, Pusan National University, Miryang 50463, Republic of Korea; prettyjiunx@naver.com (J.E.K.); hejin1544@naver.com (H.J.S.); a990609@naver.com (A.S.); dyhwang@pusan.ac.kr (D.Y.H.); 3Department of Food and Drug Biotechnology, Inje University, Gimhae 50834, Republic of Korea

**Keywords:** Indian almond (*Terminalia catappa* Linn.), lipid metabolism, inflammation, high-fat diet, obesity

## Abstract

Background: Obesity is a global health issue closely associated with dysregulated lipid metabolism and chronic inflammation. Effective strategies targeting both lipogenesis and inflammation are essential for managing obesity and its related metabolic disorders. Methods: This study evaluated the effects of *Terminalia catappa* Linn. leaf extract (TCE) on lipogenic and lipolytic pathways in high-fat diet (HFD)-induced obese mice. UPLC-QTOF-MS analysis was conducted to identify and quantify the major phenolic compounds in TCE. Mice were administered low and high doses of TCE, and various metabolic parameters, including lipid profiles, liver function markers, adipokine levels, and gene/protein expressions related to lipid metabolism and inflammation, were assessed. Results: UPLC-QTOF-MS analysis identified four major phenolic compounds in TCE—gallic acid, orientin, vitexin, and ellagic acid—with respective contents of 112.5, 163.3, 184.7, and 295.7 mg/g extract. TCE administration significantly reduced liver and adipose tissue weights, along with hepatic and adipose lipid accumulation. Both low and high doses of TCE markedly lowered serum lipid levels. Liver function was improved, as indicated by reduced levels of AST, ALT, and ALP, while BUN levels remained unchanged. On the molecular level, TCE downregulated adipogenic and lipogenic genes (PPARγ, PPARα, C/EBPα, aP2) and upregulated metabolic regulators, including leptin, adiponectin, p-HSL/HSL, and p-perilipin/perilipin, without affecting ATGL expression. TCE also suppressed pro-inflammatory cytokines such as IL-6, IL-1β, TNF-α, and TGFβ-1. Conclusions: These findings highlight the therapeutic potential of TCE in managing obesity by inhibiting lipogenesis, enhancing lipolysis, and reducing inflammation.

## 1. Introduction

According to the World Health Organization (WHO), 43% of adults aged 18 years and older were classified as overweight in 2022, with 16% affected by obesity [1,2]. These alarming statistics highlight the urgent need for effective strategies to mitigate the health risks associated with obesity [3,4,5,6]. Obesity is a complex, multisystem disease that significantly contributes to the development of metabolic disorders such as type 2 diabetes, hyperlipidemia, and atherosclerosis. It also impairs immune function, leading to chronic inflammation and increased susceptibility to infectious diseases [7,8,9,10,11].

Adipokines, particularly leptin and adiponectin, play crucial roles in the pathophysiology of obesity. Leptin levels are positively correlated with body mass index (BMI), whereas adiponectin—an important regulator of glucose and lipid homeostasis—is typically decreased in obese individuals [12]. Dysregulation of these adipokines contributes to metabolic dysfunction, including elevated serum lipid levels such as triglycerides and cholesterol. This promotes excessive fat storage in adipocytes, further aggravating conditions like hypertriglyceridemia, insulin resistance, and metabolic syndrome [13].

Natural bioactive compounds have emerged as promising agents for obesity management. Plant-derived phenolic compounds, in particular, can modulate lipid metabolism by suppressing appetite, increasing energy expenditure, and promoting lipolysis, underscoring their therapeutic potential. Our previous study demonstrated that such compounds improved lipid metabolism and oxidative stress responses in a high-sugar diet-induced Caenorhabditis elegans model [14].

*Terminalia catappa* (TC), commonly known as Indian almond, has garnered attention due to its rich phytochemical composition and diverse health benefits [13,14,15]. High-performance liquid chromatography (HPLC) analyses of TC leaf extract (TCE) have identified bioactive flavonoids including orientin, vitexin, rutin, hesperidin, and kaempferol, which are known for their anti-inflammatory, antioxidant, and metabolic regulatory properties [14,16,17]. Furthermore, in silico studies suggest that TCE may improve insulin sensitivity, facilitate glucose uptake, and inhibit the enzymatic activities of α-amylase and α-glucosidase [18].

Despite these promising findings, the molecular mechanisms underlying the anti-obesity effects of TCE remain poorly understood. Therefore, the present study aims to investigate the effects of TCE on obesity-related metabolic pathways in high-fat diet (HFD)-induced obese mice. Specifically, we evaluate its impact on lipid metabolism, lipolysis, inflammation, and key regulatory signaling molecules. These findings may provide novel insights into the therapeutic potential of TCE for the management of obesity and related metabolic disorders. These findings may provide novel insights into the therapeutic potential of TCE for the management of obesity and related metabolic disorders.

## 2. Materials and Methods

### 2.1. Plant Materials

Indian almond (*Terminalia catappa* Linn.) leaves were collected in Myanmar in 2020 and processed by the Pharmaceutical Research Department in Yangon. The harvested leaves were air-dried in the shade for approximately 7–10 days until completely dehydrated. The dried leaves were then subjected to reflux extraction using 70% (*v*/*v*) ethanol at 70 °C for 16 h. The solvent was removed under reduced pressure using a rotary evaporator (EYELA, Tokyo Rikkikai Co., Tokyo, Japan) at 60 °C. The resulting *T. catappa* leaf extract (TCE) was dissolved in dimethyl sulfoxide (DMSO) to a stock concentration of 10 mg/mL and stored at −80 °C until use. All reagents and analytical standards were molecular biology grade and obtained from Sigma-Aldrich (St. Louis, MO, USA). Solvents were chromatographic grade (Millipore, Bedford, MA, USA).

### 2.2. Purification and UPLC-Based Quantification of Phenolic Compounds

To enhance the analytical clarity of phenolic compounds in TCE, non-polar constituents were removed by partitioning the crude extract with n-hexane (1:1, *v*/*v*), followed by solid-phase extraction (SPE) for further purification. The resulting aqueous ethanol phase was loaded onto Oasis HLB cartridges (500 mg, 6 mL; Waters, Milford, MA, USA), which had been pre-conditioned with 5 mL each of methanol and distilled water. After sample loading, the cartridges were washed with 3 mL of 30% methanol to eliminate residual lipophilic impurities, and phenolic compounds were eluted with 5 mL of 80% methanol. Eluates were dried under a nitrogen stream, dissolved in methanol, and filtered (0.22 μm PTFE, Millipore, MA, USA) before analysis.

Quantitative analysis of phenolic compounds was performed using UPLC-QTOF-MS as described in our previous study. Four target analytes—gallic acid, orientin, vitexin, and ellagic acid—were selected based on prior identification. Detailed analytical conditions are provided in Appendix A.

### 2.3. Mouse Experiment Methods

All animal procedures were approved by the Institutional Animal Care and Use Committee (IACUC) of Pusan National University (Approval No. PNU-2023-0411). C57BL/6 mice were selected because they are highly susceptible to high-fat diet–induced obesity, showing pronounced weight gain, fat accumulation, and hyperglycemia, thereby serving as a well-established model for investigating obesity and metabolic disorders relevant to human dietary patterns. Four-week-old male C57BL/6 mice (Samtako BioKorea Inc., Osan, Republic of Korea) were obtained from the National Institute of Food and Drug Safety Evaluation (NIFDS, Cheongju, Republic of Korea). Mice were housed under standard conditions (23 ± 2 °C, 50 ± 10% relative humidity) with a 12 h light/dark cycle (lights on at 08:00 h and off at 20:00 h), and had ad libitum access to ultra-purified water and a sterilized irradiated chow diet (Samtako BioKorea Co., Osan, Republic of Korea), composed of 20% crude protein, 4.5% crude fat, 6.5% crude fiber, 7.5% crude ash, 0.5% calcium, and 1.0% phosphorus.

Mice were randomly assigned to dietary groups using a random number generator: a normal diet (ND) group (n = 7) and a high-fat diet (HFD) group (n = 21). HFD-fed mice were further subdivided into three groups (n = 7 each): (1) vehicle-treated group (HFD + PBS, hereafter referred to as the HFD group), (2) low-dose TCE-treated group (LTE, 100 mg/kg/day), and (3) high-dose TCE-treated group (HTE, 200 mg/kg/day). TCE was administered orally by gavage once daily at the same time each morning for 8 weeks, and to minimize order effects, the sequence of oral administration was randomized among groups each day. The sample size and dosages were determined based on prior studies using similar plant extracts [19]. All animals were included in the final analyses with no exclusion criteria. Animal husbandry and experimental procedures were conducted by different personnel to ensure blinding. At the end of the treatment period, mice were fasted for 12 h, and 24 h after the final administration, animals were euthanized by CO_2_ asphyxiation in accordance with American Veterinary Medical Association (AVMA) guidelines. Blood and tissue samples were then collected for further analyses.

### 2.4. Serum Biochemical Analysis

Blood samples were collected from the abdominal vein and transferred into serum-separating tubes (BD Vacutainer, Franklin Lakes, NJ, USA). Samples were allowed to clot at room temperature for 30 min and centrifuged at 1500× *g* for 15 min to isolate serum. Serum levels of triglycerides (TG), total cholesterol (TC), high-density lipoprotein cholesterol (HDL-C), and low-density lipoprotein cholesterol (LDL-C) were measured using a BS-120 automated chemistry analyzer (Mindray, Shenzhen, China). All biochemical assays were performed in duplicate using freshly isolated serum to ensure analytical accuracy and reproducibility. Leptin and adiponectin concentrations were quantified using a commercial ELISA kit (R&D System, Minneapolis, MN, USA).

### 2.5. Western Blot Analysis

Total protein was extracted from liver tissues using Pro-Prep Protein Extraction Solution (iNtRON Biotechnology, Gyeonggi-do, Republic of Korea) and quantified with the SMART™ BCA Protein Assay Kit (Thermo Scientific, Waltham, MA, USA). Equal amounts of protein (20–30 µg) were resolved on 4–20% SDS-polyacrylamide gels by electrophoresis at a constant voltage of 100 V for 2 h and transferred onto nitrocellulose membranes at 40 V for 2 h.

Membranes were blocked with 5% non-fat milk in TBST and incubated overnight at 4 °C with primary antibodies against p-HSL, HSL, p-Perilipin, Perilipin, ATGL, and GAPDH. After washing, membranes were incubated with HRP-conjugated secondary antibodies for 1 h at room temperature. Signals were detected using an enhanced chemiluminescence (ECL) reagent and visualized using a chemiluminescence imaging system.

Band intensities were quantified using ImageJ software (version 1.52; NIH). Phosphorylated protein levels (p-HSL and p-Perilipin) were normalized to their corresponding total protein levels, and ATGL expression was normalized to GAPDH. All data are presented as fold change relative to the normal diet group.

### 2.6. Histopathological Analysis

Liver and adipose tissues were fixed in 10% neutral-buffered formalin (pH 6.8) overnight to preserve tissue morphology and prevent autolysis. Fixed tissues were embedded in paraffin wax and sectioned into 4 µm-thick slices using a rotary microtome (Leica Microsystems, Bannockburn, IL, USA). Sections were mounted on glass slides, deparaffinized in xylene (DaeJung, Gyeonggi-do, Republic of Korea), and rehydrated through a graded ethanol series (100% to 70%) followed by distilled water.

Hematoxylin and eosin (H&E) staining was performed using reagents from Sigma-Aldrich (St. Louis, MO, USA) according to standard protocols. Histological evaluations were conducted as follows: lipid droplet accumulation in liver sections was quantified using the Leica Application Suite software (Leica Microsystems, Heerbrugg, Switzerland), and adipocyte size in adipose tissue was measured using ImageJ software (version 1.52a; NIH, Bethesda, MD, USA).

### 2.7. Quantitative Reverse Transcription Polymerase Chain Reaction (RT-qPCR) Analysis

The mRNA expression levels of PPARγ, C/EBPα, aP2, PPARα, IL-6, IL-1β, TGFβ-1, and TNF-α in liver tissues were quantified using reverse transcription-quantitative polymerase chain reaction (RT-qPCR), following previously described protocols [19]. Total RNA was isolated using RNAzol reagent (TelTest Inc., Friendswood, TX, USA), and its concentration and purity were assessed using a NanoDrop spectrophotometer (Biospecnano, Shimadzu Biotech, Kyoto, Japan).

First-strand complementary DNA (cDNA) was synthesized from 5 μg of total RNA using oligo(dT) primers, dNTPs, and SuperScript II reverse transcriptase (Invitrogen, Carlsbad, CA, USA) according to the manufacturer’s instructions. RT-qPCR was performed using gene-specific primers, 2× Power SYBR Green PCR Master Mix (Toyobo Co., Osaka, Japan), and cDNA as the template. Thermal cycling conditions were as follows: initial denaturation at 95 °C for 15 s, annealing at 55 °C for 30 s, and extension at 70 °C for 60 s. Relative gene expression levels were calculated using the comparative Ct (2^−ΔΔCt^) method, normalized to an appropriate housekeeping gene.

### 2.8. Statistical Analysis

Data were analyzed using one-way analysis of variance (ANOVA) (SPSS for Windows, version 10.10; Chicago, IL, USA), followed by Tukey’s post hoc test for multiple comparisons. Results are presented as mean ± SD.

## 3. Results

### 3.1. Phenolic Compounds in TCE

UPLC analysis identified four major phenolic compounds in TCE: gallic acid, orientin, vitexin, and ellagic acid (Figure 1A). The corresponding molecular ion peaks of these compounds were confirmed by UPLC-QTOF-MS, as shown in Appendix A. Quantitative analysis using authentic standards revealed that the contents of gallic acid, orientin, vitexin, and ellagic acid were 112.5 mg/g, 163.3 mg/g, 184.7 mg/g, and 295.7 mg/g extract, respectively (Figure 1B).

### 3.2. Effects of TCE on Body Weight in HFD-Fed Mice

To assess the effects of TCE on body weight regulation and consumption behavior, mice were administered TCE daily for 8 weeks, during which body weight, food intake, and water consumption were continuously monitored (Figure 2A–C). At the end of the treatment period, both LTE and HTE groups showed significantly lower body weights compared to the HFD group, exhibiting a clear dose-dependent effect. However, the body weights of TCE-treated mice remained slightly higher than those of the ND group (Figure 2B). No significant differences in food or water intake were observed among the groups (Figure 2C), suggesting that the reduction in body weight was not attributable to altered consumption.

### 3.3. Effects of TCE on Serum Lipid and Liver Enzyme Profiles in HFD-Fed Mice

To assess the effects of TCE on lipid metabolism, serum lipid levels were measured in HFD-fed mice (Figure 3A–D). HFD feeding significantly increased TC, TG, HDL, and LDL levels by approximately 40–70% compared to the ND group. Both low-dose and high-dose TCE treatments (LTE and HTE, respectively) significantly reduced these lipid parameters relative to the HFD group, indicating improved lipid regulation.

Hepatic and renal function markers, including ALT, AST, ALP, and BUN, were evaluated (Figure 3E–H). ALT and AST levels were significantly elevated in HFD-fed mice, reflecting hepatic stress caused by lipid accumulation. No significant differences were observed in ALP levels among HFD groups. TCE administration induced a dose-dependent decrease in ALT and AST, suggesting hepatoprotective effects. BUN levels remained stable across all groups, indicating no adverse effects of TCE on renal function.

### 3.4. TCE Reduces Abdominal Fat Accumulation and Adipocyte Hypertrophy in HFD-Fed Mice

Abdominal fat mass was significantly elevated in the HFD group compared to the ND group. While the LTE group did not exhibit a significant reduction relative to the HFD group, the HTE group showed a marked decrease in fat mass (*p* ≤ 0.05) (Figure 4A,B).

Histological analysis further revealed a substantial enlargement of adipocyte area in the HFD group relative to the ND group. TCE treatment resulted in a significant, dose-dependent reduction in adipocyte size (Figure 4A,C). Compared to the ND group, adipocyte area was approximately threefold higher in the HFD group, while the LTE and HTE groups demonstrated reductions of approximately 50% and 70%, respectively. These findings suggest that TCE effectively attenuates adipocyte hypertrophy in a dose-responsive manner (Figure 4C).

### 3.5. TCE Reduces Liver Weight and Hepatic Lipid Accumulation in HFD-Fed Mice

As shown in Figure 4, liver weights were elevated in the HFD group compared to the ND group. Treatment with both LTE and HTE resulted in a significant reduction in liver weight relative to HFD, although the reduction did not display a clear dose-dependent pattern (Figure 5A,B).

Histological analysis of liver sections revealed a marked increase in the number of lipid droplets per unit area in the HFD (>400 droplets/mm^2^) compared to the ND (<20 droplets/mm^2^). Both LTE and HTE groups showed a significant, dose-dependent decrease in lipid droplet number, with LTE reducing the lipid droplets by approximately 50% relative to HFD, and HTE inducing a markedly greater reduction, although not to the level of the ND group (Figure 5A,C). These findings suggest that TCE effectively attenuates hepatic lipid accumulation in HFD-fed mice.

### 3.6. TCE Regulates Hepatic Lipid Metabolism by Modulating Lipogenesis and Lipolysis in HFD-Fed Mice

To elucidate the molecular mechanisms underlying the hypolipidemic effects of TCE, hepatic mRNA expression of key genes involved in adipogenesis and lipogenesis was evaluated. Compared to the ND group, the HFD group exhibited significantly upregulated expression of PPARα, PPARγ, C/EBPα, and aP2. Treatment with TCE at both LTE and HTE doses induced downregulation of PPARα and PPARγ mRNA levels (Figure 6A,B). C/EBPα expression was reduced in the LTE group but increased in the HTE group, approaching levels observed in the HFD group (Figure 6C). Expression of aP2, a marker of adipocyte differentiation, was significantly decreased in TCE-treated groups compared to the HFD in a dose-dependent manner (Figure 6D). These results indicate that TCE modulates genes related to lipid metabolism, contributing to its regulatory effects on hepatic lipid accumulation.

To further investigate the effect of TCE on lipolysis, protein expression levels of key lipolytic regulators were assessed. The ratio of phosphorylated HSL to total HSL (p-HSL/HSL) was significantly decreased in the HFD group compared to the ND group, indicating suppressed lipolytic activity. TCE administration restored phosphorylated HSL (p-HSL)/HSL levels in a dose-dependent manner, suggesting enhanced lipolysis upon treatment (Figure 6E). In contrast, phosphorylated Perilipin (p-Perilipin)/Perilipin ratios were elevated in the HFD group and further increased in both LTE and HTE groups, albeit without a clear dose–response relationship (Figure 6F). ATGL protein expression remained unaltered across all groups (Figure 6G), suggesting that the lipolytic effect of TCE may not involve ATGL modulation.

### 3.7. TCE Mitigates Inflammation by Modulating Pro-Inflammatory Cytokines in HFD-Fed Mice

To assess the inflammatory response, the relative mRNA expression levels of key pro-inflammatory cytokines—IL-6, TGF-β1, IL-1β, and TNF-α—were measured (Figure 7). Compared to the ND group, the HFD group showed a significant upregulation of these cytokines. TCE treatment led to a dose-dependent decrease in their expression. Notably, the mRNA levels of IL-6 and TNF-α were restored to levels comparable to those in the ND group following TCE administration. These results suggest that TCE effectively alleviates HFD-induced hepatic inflammation by modulating pro-inflammatory cytokine expression.

### 3.8. TCE Restores Leptin and Adiponectin Expression in HFD-Fed Mice

Leptin and adiponectin are adipokines that regulate energy balance and metabolism; leptin controls appetite and energy use, while adiponectin improves insulin sensitivity and reduces inflammation.

In the HFD group, mRNA expression levels of both leptin and adiponectin were significantly reduced compared to the ND group, indicating metabolic dysregulation induced by the HFD. TCE administration resulted in a dose-dependent increase in the expression of both adipokines (Figure 8), suggesting that TCE may improve metabolic function by modulating leptin and adiponectin expression in the liver.

## 4. Discussion

In this study, we refined the profiling method of phenolic compounds in TCE by introducing a SPE step before UPLC-QTOF-MS analysis. Compared to our previous study, four major phenolic compounds—gallic acid, orientin, vitexin, and ellagic acid—were consistently identified, reaffirming their predominance in TCE [14]. Notably, while the previous study focused on qualitative analysis, this study employed quantitative analysis. As a result, the contents of gallic acid, orientin, vitexin, and ellagic acid were determined to be 112.5 mg/g, 163.3 mg/g, 184.7 mg/g, and 295.7 mg/g extract, respectively. The total amount of these compounds was approximately 800 mg/g extract, which is higher than the previously measured total phenolic content (TPC) of 700 mg GAE/g extract. This increase is attributable to the SPE step, which selectively removed non-polar matrix interferences, leading to a cleaner extract and improved chromatographic resolution. SPE is widely employed in phytochemical analysis to enhance the detection sensitivity and accuracy of target analytes by minimizing matrix effects and concentrating polar compounds [20,21]. As a result, the chromatographic baseline was markedly stabilized, facilitating more precise quantification of phenolic constituents in TCE.

The association between high-fat diet (HFD) consumption and weight gain is well established [22,23], and our study provides further evidence supporting the efficacy of natural interventions, such as TCE, in mitigating diet-induced obesity. Given the multifactorial nature of obesity—including genetic predisposition, stress, metabolic regulation, and physical activity—therapeutic strategies beyond caloric restriction are needed [24,25]. The significant body weight reduction observed in TCE-treated groups suggests potential metabolic benefits, despite no differences in food and water intake across groups. This implies that TCE may enhance energy expenditure or suppress adipose tissue accumulation.

White adipose tissue (WAT) expansion during obesity is primarily driven by adipocyte hypertrophy and hyperplasia [26,27]. In this study, TCE dose-dependently reduced both abdominal and hepatic fat accumulation. The decrease in hepatic lipid droplets and adipocyte size further supports the anti-steatotic properties of TCE. These lipid-lowering and anti-adipogenic effects are consistent with previous reports involving polyphenol-rich plant extracts [28,29,30,31]. Collectively, the data suggest that TCE may act by suppressing lipogenesis and triglyceride accumulation in hepatocytes, indicating its therapeutic potential in fatty liver disease and HFD-induced hepatic dysfunction.

Consistent with known links between dyslipidemia and cardiometabolic diseases [32], HFD significantly increased serum TC, TG, LDL, and HDL levels. TCE administration attenuated these changes, suggesting a hypolipidemic effect mediated by its bioactive components. Importantly, reductions in hepatic and abdominal fat mass were accompanied by improvements in serum lipid profiles, as similarly reported by Tijjani et al. [33], supporting the role of plant-derived compounds in managing both tissue lipid deposition and circulating lipid levels.

Markers of hepatic function—ALT, AST, and ALP—are typically elevated in response to liver injury or metabolic stress [34,35]. In our study, ALT and AST levels were significantly elevated in HFD-fed mice, reflecting hepatic dysfunction. TCE treatment effectively reduced these enzyme levels in a dose-dependent manner, suggesting a hepatoprotective effect, likely via anti-inflammatory or antioxidant mechanisms. While ALP levels were reduced across all experimental groups, the absence of a dose-specific trend suggests it may be less responsive to the hepatic effects of TCE. In addition, BUN levels remained unchanged, indicating that TCE did not adversely affect renal function.

Lipid metabolism is governed by the balance between lipogenesis, adipogenesis, and lipolysis. HFD intake significantly upregulated hepatic expression of PPARγ, PPARα, C/EBPα, and aP2—key transcription factors involved in lipid synthesis and adipocyte differentiation—consistent with earlier studies [34,35]. TCE administration markedly downregulated PPARγ and PPARα, suggesting suppression of lipid accumulation pathways. While phytochemicals such as curcumin have been reported to upregulate these genes [36,37], others like phlorotannins and apigenin exhibit similar inhibitory effects to those seen with TCE [38,39], possibly due to differences in compound specificity or tissue response [40,41].

Interestingly, C/EBPα expression showed a dose-specific response: it was downregulated by LTE but upregulated by HTE. This pattern may reflect concentration-dependent modulation of distinct metabolic and inflammatory pathways, consistent with previous findings [42,43]. Additionally, the suppression of aP2—implicated in lipid transport and storage—supports the anti-adipogenic effects of TCE.

On the lipolytic front, TCE treatment increased phosphorylation of HSL and perilipin, markers of triglyceride hydrolysis and fatty acid mobilization [44,45]. Although ATGL expression remained unchanged, the enhanced phosphorylation of HSL and perilipin suggests activation of hepatic lipolysis. As perilipin phosphorylation facilitates HSL translocation to lipid droplets, their concurrent upregulation indicates coordinated promotion of lipolytic activity. This mechanism aligns with established models where perilipin-mediated access to lipid stores enhances lipid mobilization and energy regulation [46,47].

Taken together, these findings suggest that TCE reduces hepatic lipid accumulation by suppressing lipogenesis-related gene expression and promoting lipolysis, highlighting its potential as a natural hypolipidemic agent.

Obesity-induced hepatic inflammation is closely linked to adipokine imbalance and pro-inflammatory cytokine production. Leptin and adiponectin, two key adipokines, play crucial roles in hepatic lipid metabolism and immune modulation [48]. In this study, HFD significantly reduced leptin and adiponectin levels, reflecting impaired adipokine signaling and heightened hepatic inflammation. TCE administration restored leptin levels, particularly in the LTE group, and increased adiponectin in both TCE-treated groups, suggesting improved metabolic recovery.

Pro-inflammatory cytokines including IL-1β, IL-6, TNF-α, and TGF-β1 were markedly elevated in the HFD group, consistent with hepatic inflammation likely driven by lipotoxicity and ceramide accumulation [46]. TCE significantly reduced the expression of these cytokines, with a dose-dependent reduction observed for IL-6, IL-1β, and TGF-β1. Although TNF-α did not follow a clear dose–response trend, its overall decline underscores the anti-inflammatory potential of TCE. These effects are likely mediated through inhibition of key inflammatory pathways such as NF-κB and JNK, as supported by prior studies in *C. elegans* and other models [14,49].

While these findings collectively highlight the therapeutic potential of TCE in ameliorating diet-induced obesity and hepatic dysfunction, it remains to be elucidated which specific phenolic compounds are primarily responsible for these bioactivities. Although gallic acid, orientin, vitexin, and ellagic acid are predominant in TCE, the individual or synergistic contributions of these compounds to metabolic and anti-inflammatory effects of TCE require further investigation. Therefore, future studies focusing on isolating and evaluating the bioactivity of each constituent are warranted to delineate the precise active components responsible for therapeutic effects of TCE. In addition, although the restoration of leptin and adiponectin was demonstrated, further studies are needed to clarify the involvement of specific metabolic pathways, such as AMPK, PPAR, and insulin signaling, in mediating these effects. Finally, the present study was conducted exclusively in C57BL/6 mice; comparative investigations using other rodent strains commonly applied in obesity research will be essential to validate the generalizability of our findings.

## 5. Conclusions

This study demonstrates that TCE exerts beneficial effects against HFD-induced obesity and hepatic steatosis in mice. TCE administration improved metabolic parameters, including body weight, lipid profiles, liver enzyme levels, and hepatic histology, in a dose-dependent manner. Mechanistically, TCE downregulated lipogenic and adipogenic genes (PPARγ, PPARα, C/EBPα, aP2) while enhancing lipolytic activity via increased phosphorylation of HSL and perilipin. TCE also reduced pro-inflammatory cytokines and restored adipokines such as leptin and adiponectin, suggesting coordinated anti-steatotic and anti-inflammatory actions.

Overall, these findings support the therapeutic potential of TCE as a nutraceutical or phytopharmaceutical for managing obesity-related hepatic and metabolic disorders. However, further studies are needed to determine which phenolic constituents, such as gallic acid, orientin, vitexin, and ellagic acid, are primarily responsible for these effects.

## Figures and Tables

**Figure 1 metabolites-15-00594-f001:**
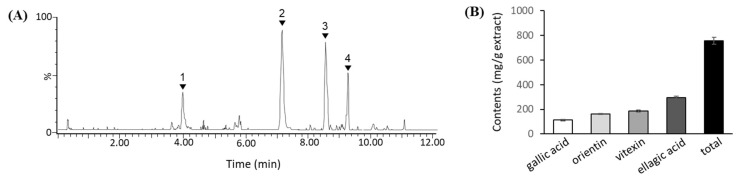
Identification and quantification of phenolic compounds in TCE. (**A**) Representative UPLC chromatogram of phenolic compounds detected in TCE (peak 1: gallic acid, peak 2: orientin, peak 3: vitexin, peak 4: ellagic acid). (**B**) Quantitative analysis of gallic acid, orientin, vitexin, and ellagic acid in TCE. Data are expressed as mg/g extract.

**Figure 2 metabolites-15-00594-f002:**
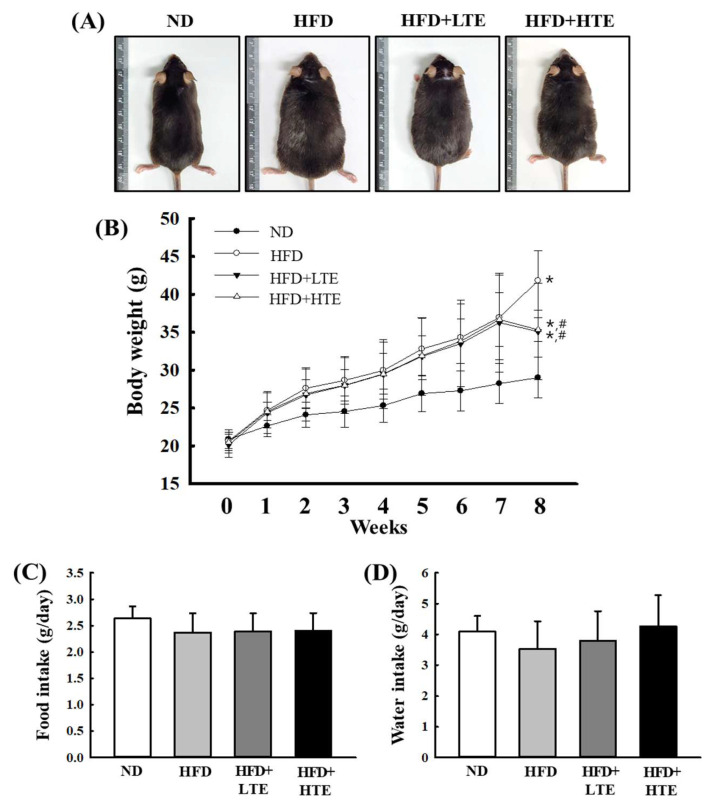
Effects of TCE on Body Weight and Consumption Behavior in HFD-Fed Mice. (**A**) Representative images of mice following 8 weeks of treatment under each dietary condition. (**B**) Weekly body weight progression over the treatment period. (**C**) Average daily food intake, and (**D**) average daily water intake across groups. Data are expressed as mean ± SEM from three independent determinations. * and #: significant difference compared to the ND and HFD, respectively (*p* ≤ 0.05).

**Figure 3 metabolites-15-00594-f003:**
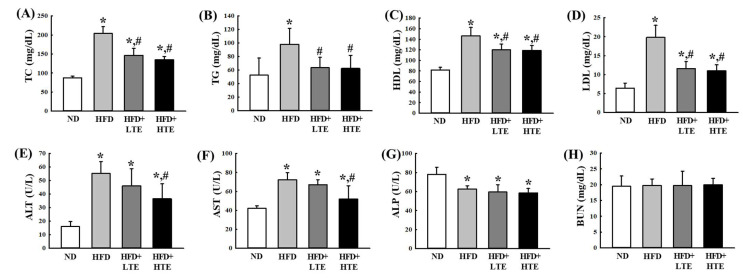
Effects of TCE on serum lipid levels and liver enzymes in HFD-fed mice. (**A**) TC, (**B**) TG, (**C**) HDL, (**D**) LDL, (**E**) ALT, (**F**) AST, (**G**) ALP, and (**H**) BUN. Mice were fed a normal diet (ND) or high-fat diet (HFD) for 8 weeks, during which TCE was orally administered once daily at 100 mg/kg (LTE) or 200 mg/kg (HTE). Data are expressed as mean ± SEM from three independent determinations. * and #: significant difference compared to the ND and HFD, respectively (*p* ≤ 0.05).

**Figure 4 metabolites-15-00594-f004:**
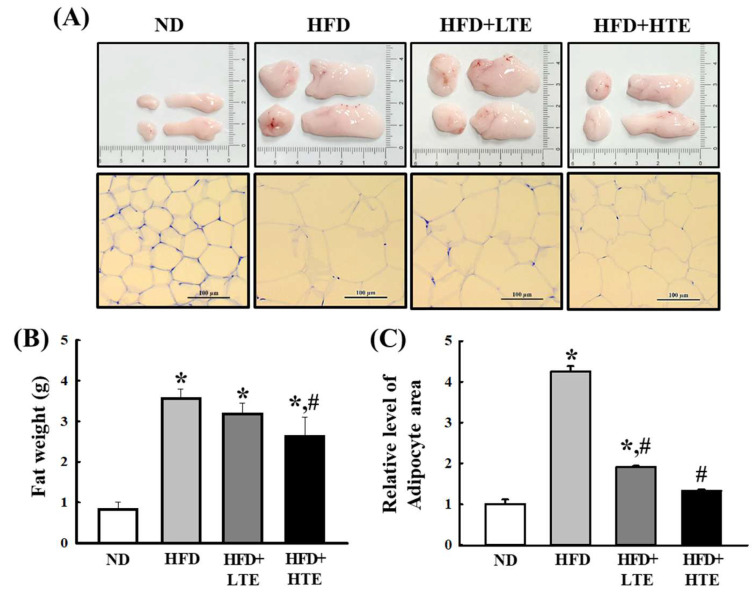
Effects of TCE on abdominal fat mass and adipocyte hypertrophy in HFD-fed mice. Mice were fed a normal diet (ND) or high-fat diet (HFD) for 8 weeks, during which TCE was orally administered once daily at 100 mg/kg (LTE) or 200 mg/kg (HTE). (**A**) Representative images of abdominal fat pads and H&E-stained adipose tissue sections from each group (ND, HFD, HFD + LTE, and HFD + HTE). (**B**) Quantification of abdominal fat weight. (**C**) Relative adipocyte area measured from histological sections. Data are expressed as mean ± SEM from three independent determinations. * and #: significant difference compared to the ND and HFD, respectively (*p* ≤ 0.05).

**Figure 5 metabolites-15-00594-f005:**
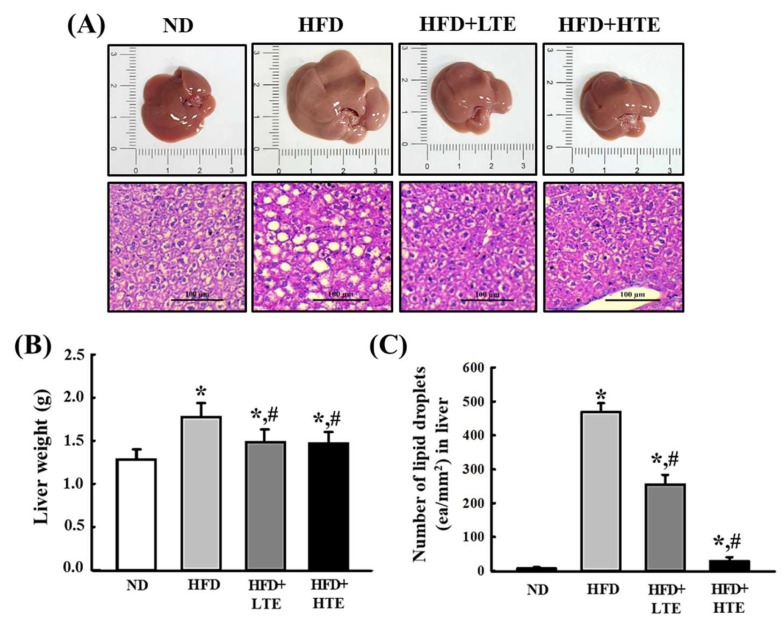
Effects of TCE on Liver Weight and Hepatic Lipid Droplet Accumulation in HFD-Fed Mice. Mice were fed a normal diet (ND) or high-fat diet (HFD) for 8 weeks, during which TCE was orally administered once daily at 100 mg/kg (LTE) or 200 mg/kg (HTE). (**A**) Representative images of liver tissue and histological sections showing lipid droplets from each group (ND, HFD, HFD + LTE, and HFD + HTE). (**B**) Quantification of liver weights. (**C**) Number of lipid droplets per unit area in liver sections. Data are expressed as mean ± SEM from three independent determinations. * and #: significant difference compared to the ND and HFD, respectively (*p* ≤ 0.05).

**Figure 6 metabolites-15-00594-f006:**
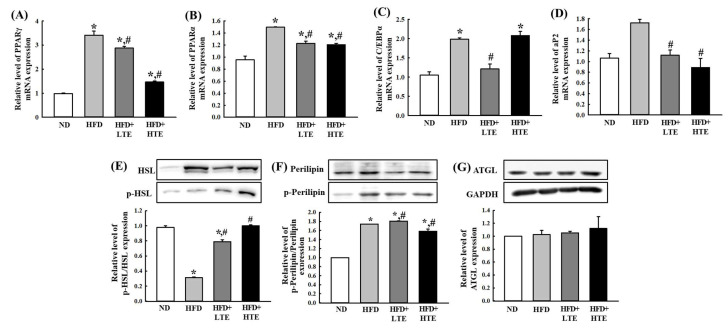
Effects of TCE on hepatic mRNA expression of genes involved in adipogenesis and lipogenesis, and on protein expression of lipolysis markers. Mice were fed a normal diet (ND) or high-fat diet (HFD) for 8 weeks, during which TCE was orally administered once daily at 100 mg/kg (LTE) or 200 mg/kg (THE). Hepatic mRNA levels of (**A**) PPARγ, (**B**) PPARα, (**C**) C/EBPα, and (**D**) aP2 were analyzed by qRT-PCR. Hepatic protein levels of (**E**) phosphorylated HSL relative to total HSL (p-HSL/HSL), (**F**) phosphorylated Perilipin relative to total Perilipin (p-Perilipin/Perilipin), and (**G**) ATGL were analyzed by Western blotting. Data are expressed as mean ± SEM from three independent determinations. * and #: significant difference compared to the ND and HFD, respectively (*p* ≤ 0.05).

**Figure 7 metabolites-15-00594-f007:**
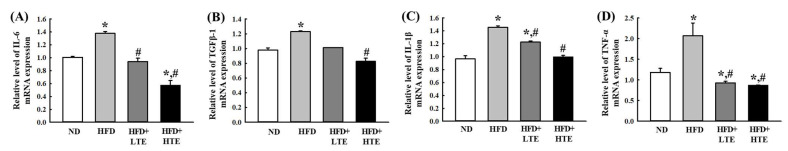
Effects of TCE on hepatic mRNA expression of pro-inflammatory cytokines. Mice were fed a normal diet (ND) or high-fat diet (HFD) for 8 weeks, during which TCE was orally administered once daily at 100 mg/kg (LTE) or 200 mg/kg (THE). Hepatic mRNA levels of (**A**) IL-6, (**B**) TGF-β1, (**C**) IL-1β, and (**D**) TNF-α were analyzed by qRT-PCR. Data are expressed as mean ± SEM from three independent determinations. * and #: significant difference compared to the ND and HFD, respectively (*p* ≤ 0.05).

**Figure 8 metabolites-15-00594-f008:**
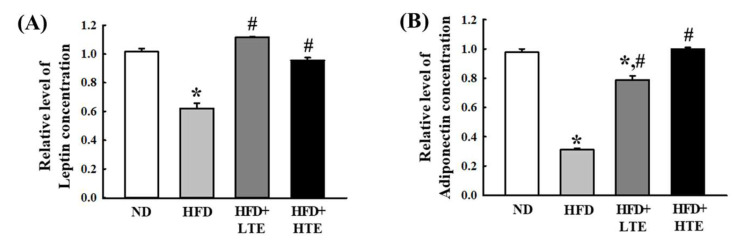
Effects of TCE on hepatic leptin and adiponectin expression in HFD-fed mice. (**A**) Leptin, and (**B**) adiponectin. Mice were fed a normal diet (ND) or high-fat diet (HFD) for 8 weeks, during which TCE was orally administered once daily at 100 mg/kg (LTE) or 200 mg/kg (THE). Protein levels of leptin and adiponectin were measured by ELISA. Data are expressed as mean ± SEM from three independent determinations. * and #: significant difference compared to the ND and HFD, respectively (*p* ≤ 0.05).

## Data Availability

The data that support the findings of this study are available from the corresponding author upon reasonable request. Data sharing is restricted at this time as the results are subject to pending intellectual property rights and patent considerations.

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
