# Peer review of "Phenolic-Rich Indian Almond (Terminalia catappa Linn) Leaf Extract Ameliorates Lipid Metabolism and Inflammation in High-Fat Diet (HFD)-Induced Obese Mice"

_metabolites, 2025, doi:10.3390/metabo15090594_

Round 1

Reviewer 1 Report

Comments and Suggestions for Authors

Dear Authors,
I hope this message finds you well.

Thank you for the opportunity to review the manuscript: “Phenolic-rich Indian almond (Terminalia catappa Linn) leaf extract ameliorates lipid metabolism and inflammation in high-fat diet (HFD)-induced obese mice.”

Please find below my review with recommendations and suggestions.

Sincerely,

Author Response

We sincerely appreciate the reviewer’s valuable comments.
In response, 
1) we have added the Introduction to emphasize the clinical significance of our study, “These findings may provide novel insights into the therapeutic potential of TCE for the management of obesity and related metabolic disorders,”. This adjustment enhances the flow and presents the objective more concisely and directly.
2) In accordance with the ARRIVE 2.0 guidelines, we have revised the Methods section (2.3 Mouse Experiment Methods) to provide additional details. Specifically, we have included information on randomization procedure, inclusion and exclusion criteria, blinding between husbandry and experimental procedures, and further details on euthanasia following AVMA guidelines. These revisions strengthen the transparency and reproducibility of our animal study.
3) Regarding the Discussion, we fully agree with the reviewer’s suggestion. We have now included a statement proposing further investigation into the specific metabolic pathways, including AMPK, PPAR, and insulin signaling, that may underlie the restoration of leptin and adiponectin. Furthermore, we have acknowledged the limitation of using only C57BL/6 mice and suggested comparative studies with other rodent strains to enhance the generalizability of our findings.

We believe that these revisions and additions have improved the clarity, rigor, and scientific value of the manuscript.

Reviewer 2 Report

Comments and Suggestions for Authors

The work is interesting, but I have a few comments:

  1. Why was CO2 used for euthanasia in the animal model?
  2. Are the obtained results reliable, as CO2 reacts easily?
  3. Therefore, I have doubts about the evaluation of phenolic compounds such as gallic acid, orientin, vitexin, and ellagic acid?
  4. How long were the mice euthanized with CO2?
  5. Is there no description of the production of interleukins and adipokines in the Materials and Methods section?
  6. The obtained results require confirmation of the therapeutic potential of TCE as a nutraceutical or phytopharmaceutical in the treatment of obesity-related liver and metabolic disorders, but the mice should be euthanized using a different method.

Author Response

We sincerely appreciate the reviewer’s valuable comments.

1) COâ‚‚ euthanasia was employed in this study in accordance with the IACUC-approved protocol and the AVMA (2020) guidelines. This method is widely accepted as humane and does not chemically interfere with tissue metabolites such as phenolic compounds. We revised the Methods section to include details.

2) We would like to clarify that the description of interleukin measurements is already included in the Materials and Methods section. However, we recognized that the methodological details for leptin and adiponectin analyses were not clearly described in the original submission. In the revised manuscript, we have now added the procedures for leptin and adiponectin measurement in the Materials and Methods section to provide a more complete description of the experimental design.
